# Luminescence Reduced Graphene Oxide Based Photothermal Purification of Seawater for Drinkable Purpose

**DOI:** 10.3390/nano12101622

**Published:** 2022-05-10

**Authors:** Jin Huang, Zhen Chu, Christina Xing, Wenting Li, Zhongxin Liu, Wei Chen

**Affiliations:** 1Key Laboratory of Advanced Materials of Tropical Island Resources of Ministry of Education, School of Materials and Chemical Engineering, Hainan University, Haikou 570228, China; 68828813@hainanu.edu.cn (J.H.); 19085216210007@hainanu.edu.cn (Z.C.); 2Hainan Provincial Key Lab of Fine Chem, School of Materials and Chemical Engineering, Hainan University, Haikou 570228, China; 3Department of Physics, The University of Texas at Arlington, Arlington, TX 76019-0059, USA; christina.xing@uta.edu; 4Key Laboratory of Functional Molecular Engineering of Guangdong Province, School of Chemistry and Chemical Engineering, South China University of Technology, Guangzhou 510641, China; wenting@scut.edu.cn

**Keywords:** graphene oxide nonwoven membrane, solar evaporation, air–water interface, water purification, photothermal conversion, cell viability, cytotoxicity

## Abstract

Getting drinking water from seawater is a hope and long-term goal that has long been explored. Here, we report graphene-loaded nonwoven fabric membranes for seawater purification based on photothermal heating. The photothermal membrane of non-woven fabric loaded with graphene oxide has high light absorption and strong heating effect, and its evaporation rate about 5 times higher than that of non-woven fabric. Under the condition of light intensity of 1 kW m^−2^, the evaporation rate can reach 1.33 kg m^−2^ h^−1^. The results of cell activity test showed that the concentration of bacteria after photothermal membrane treatment decreased significantly. The photothermal membrane can be used for many times without greatly reducing the evaporation efficiency, which means that it is suitable for regional water purification and seawater desalination.

## 1. Introduction

The shortage of safe drinking water is one of the most severe global challenges facing human society. Desalination is one of the most cost-effective ways to increase drinking water supply. In recent years, the development of seawater desalination industry has made great progress. For example, as one of the most widely used desalination technologies, reverse osmosis (RO) has a recovery rate of about 50% and a consumption of only 2 kw·h·m^−3^. However, due to the need for complex infrastructure and high energy consumption, this technology is still unavailable in most underdeveloped countries. In addition, with the increase of feed water salinity, the working pressure, scaling probability and fuel consumption of the reverse osmosis system increase significantly [1,2,3,4]. Therefore, the development of new green seawater desalination technology is particularly important. At present, solar powered desalination is becoming one of the most promising technologies to increase the supply of clean water, because of the abundance of solar energy and seawater and their negligible carbon footprint, especially in remote areas lacking electricity or infrastructure [5,6,7,8].

To promote the further development of direct solar desalination, great efforts have been made to improve the efficiency of solar thermal conversion, prolong the service life of equipment and reduce the cost of solar desalination [9,10,11,12,13]. Some studies show that the evaporator with optimized structure design can produce fresh water and collect the by-product salt in salt water at the same time, to achieve zero liquid discharge. To strengthen the relationship between water and energy and alleviate the energy shortage, various solar powered hydropower hydrogen production systems have also attracted people’s attention. This system converts waste heat into electric energy, and then drives electrochemical decomposition of water, which not only improves the overall energy conversion efficiency, but also promotes the production of green fuel [14,15]. According to the mechanism of photothermal conversion, photothermal materials are mainly divided into three categories, such as plasma metal nanoparticles, semiconductors and carbon-based materials. Among them, carbon-based materials are ideal for solar evaporative power generation because of their excellent thermal properties, low cost and abundant source materials [16,17]. As an important carbon-based material, graphene oxide (Go) or reduced graphene oxide (rGO) has thin nanostructures, large surface area, low molar specific heat, stable mechanical strength and excellent light absorption covering the whole solar spectral range (250–2500 nm). Therefore, they are directly used as light absorbers or photothermal layers for various substrates (such as wood, sponge, polymer membrane and natural fiber) [18,19,20].

To better transfer heat to the water body that needs evaporation, good heat management performance and excellent light absorbing materials are two key factors. For the heat management strategy, the traditional self-floating evaporation configuration generally has large heat loss due to the direct contact of the light absorbing material with the water body. In contrast, the use of thermal insulator assisted evaporation device can minimize the heat and conduction loss of photothermal materials [21,22]. At the same time, interface heating can limit the absorbed heat energy to a small amount of interface water in the upper layer, to shorten the start-up time of steam generation and increase the water evaporation rate. Although some transport auxiliary evaporators have been reported, their evaporation performance is still hindered due to the lack of good design structure. Therefore, the evaporation material must have a highly porous structure and hydrophilic channels to promote the transmission of water and the escape of water vapor. Non-woven fabric is made of polyester fiber and polyester fiber (PET for short). It has the characteristics of antibacterial, alkali corrosion and strong hydrophilicity [23,24]. In addition, Yu and others think that the hydrogel polymer network can restrict the water mass in the molecular grid, effectively reduce the enthalpy of vaporization, which is beneficial to improve the evaporation performance of the self-floating gel evaporator [25,26,27]. However, the combination of nanofiber hydrogel and rGO as a feasible strategy to improve the evaporative performance of the transport assisted evaporator and the mechanism of controlling the interfacial evaporation need further study.

Here, we report an approach based on reduced graphene oxide (rGO)-loaded nonwoven membrane for localizing evaporation, which is cost-effective and environmentally friendly. The hydrophilic non-woven transports water to the hot region by capillary forces. The rGO non-woven composite membrane has a superior mechanical stability. This method can be used repeatedly for many times. Under 1 sun intensity (1 kW m^−2^), the evaporation rate of photothermal membrane reached 1.33 kg·m^−2^·h^−1^**.** In addition, we tested the cell viability on the original seawater and the purified water. The results show the purified water has no toxicity, while the seawater is toxic to cells at a certain concentration. This indicates this is an excellent method for purification of drinkable water from seawater. The simple design of the system, along with the low costs, makes it possible for practical applications as an outstanding solution to the long-term challenge of drinking water issues.

## 2. Experiment Section

### 2.1. Materials

Graphite powder is purchased from Tianjin Dengke Chemical Reagent Co, Ltd. (Tianjin, China), H_2_SO_4_, KMnO_4_ and H_2_O_2_ are obtained from Aldrich, Deionized water, self-made in the laboratory, non-woven fabric is provided by Hainan Xinglong Nonwoven Company (Hainan, China).

### 2.2. Preparation of Graphene Oxide (Go)

GO is synthesized by the modified Hummers methods [28,29,30]. Graphite (1 g) are mixed in 23 mL H_2_SO_4_, and the mixture is stirred for 2 min in an ice bath at a temperature less than 20 °C. Then, add 3 g KMnO_4_ and stir for 3 h. The ice bath is then removed, and the mixture is stirred at 35 °C for 8 h. Next, 100 mL water is dropped gently. It is then diluted with 50 mL water and 5 mL H_2_O_2_. The mixture is washed with HCl (5%) and deionized (DI) water and centrifugation, freeze-drying to obtain samples.

### 2.3. Preparation of rGO-Based Nonwoven Fabric Membrane (RGM)

First, 0.05 g graphite oxide be dispersed in 10 mL DI water by sonication for 1 h to make an aqueous dispersion. Cut a round non-woven fabric into graphene oxide solution, taking it out, and by centrifugation, removing the excess graphene oxide, and cycling it several times. Then, the non-woven membrane loaded with GO was placed in an oven at 180 °C and reacted for 5 h to obtain RGM photothermal membrane [31].

### 2.4. Characterization

The structural analysis of graphene oxide was characterized 31 by power X-ray diffraction using a Bruker D8-advance. The Raman spectra were obtained by a confocal laser Raman spectrometer. The size and morphology were obtained by scanning electron microscopy (SEM, Hitachi S-4800). The transmission electron microscope (TEM) was characterized by a JEOL JEM 2100. The real-time temperatures of the samples were measured by an IR thermography (HT-02, Xin Site China). The 980 nm IR laser with tunable powers was generated by a diode laser system (BWT, Beijing, China).

### 2.5. Cell Toxicity Studies by MTT (3-(4,5-Dimethylthiazol-2-yl)-2,5-diphenyltetrazolium Bromide) Assay

The cytotoxicity of the seawater and the purified water were evaluated by means of MTT assay on Hela Cells. Hela Cells-CCL-2 were purchased from ATCC American Type Culture Collection. Cells were cultured in high-glucose Dulbecco’s Modified Eagle’s Medium (H-DMEM) containing 10% FBS and 1% penicillin-streptomycin at 37 °C in a humidified environment containing 5% CO_2_. Before the experiment, the cells were pre-cultured until confluence was reached. MTT assay is a colorimetric assay which is in widespread use for assaying cell proliferation and cytotoxicity [32]. First, 10,000 cells/well were seeded in 96 well plates and incubated for 24 h for the completion of cell attachment. Cells were divided into three groups: seawater, purified water and control (untreated) groups. Then, the plates were incubated for another 24 h to allow cells to uptake water. MTT assay was prepared by diluting 5 mg/mL stock solution with media by a factor of 10. Then, 100 µL of MTT assay was added to each well, replacing old media. The 96 well plates were then incubated for 3 h at 37 °C under humidified atmosphere. After incubation for 3 h, MTT solution was removed and 100 µL DMSO (dimethyl sulfoxide) was added to solubilize formazan crystal. The formazan crystal becomes purple-colored with DMSO dissolution. The viability of cells was directly dependent on the absorption of formazan solution.

After adding water, 96 well plates be incubated for 3 h, then 0.5 mg/mL MTT solution was added to each well and incubated for 3 h. Yellow colored MTT converted into purple colored formazan crystal. Formazan is insoluble in aqueous solution. Formazan is solubilized by adding 100 Ul DMSO to each well. The absorbance of formazan is directly proportional live cell counts and can be employed to present a relative cell viability as compared to control. The absorbance of Formazan solution was measured using a multiskan FC microplate photometer (Fisher Scientific, Hampton, NH, USA) at 540 nm.

Cell viability was calculated as follows:$$\mathrm{Cell}\;\mathrm{viability}=\frac{\mathrm{The}\;\mathrm{absorbance}\;\mathrm{of}\;\mathrm{the}\;\mathrm{treatment}\;\mathrm{group}}{\mathrm{The}\;\mathrm{absorbance}\;\mathrm{of}\;\mathrm{the}\;\mathrm{control}\;\mathrm{group}}\times 100\%$$

## 3. Results and Discussion

### 3.1. Preparation of the Graphene Oxide-Loaded Hydrophilic Nonwoven

Figure 1 shows the transmission electron microscope (TEM) images of the prepared GO with average size of ~100 nm, in which the GO exhibits lamellar structure with some laminations slightly stacked. There is a slight sliding between the graphene oxide layered structures. The magnified TEM image on the right shows that the microscopic size of the entire sheet is nanoscale.

Figure 2 is the Raman spectrum of GO. The D peak and the G peak are Raman characteristic peaks of carbon atom crystals, respectively, and their diffraction peaks are centered at 1300 cm^−1^ and 1580 cm^−1^, respectively. D peak indicates some defects of the carbon atom lattice, and the G peak indicates the in-plane stretching vibration state of the sp^2^ hybridization of carbon atoms. The D peak and G peak of GO independently developed by the laboratory are at 1359 cm^−1^ and 1615 cm^−1^, respectively. *I*_D_/*I*_G_ is the intensity ratio of D peak to G peak. This ratio can refer to the intensity relationship between the two peaks. In the Figure 2, *I*_D_/*I*_G_ is about 0.8, which is less than the generally reported literature value [33,34], indicating that the properties of the synthesized GO are closer to the characteristics of graphene oxide, the fewer the defects on its surface and the higher the chemical stability, which will help to improve the stability and service life of the photothermal membrane.

As shown in Figure 3, there is a sharp diffraction peak at 9.96°, slightly lower than the characteristic peak of standard GO centered at about 11° [5,35]. This may be due to the intercalation of H^+^ in the go layer during the stripping preparation of GO, which increases the distance between the crystal planes of GO. This expansion layer can aggravate the exposure rate of the edge and improve the overall light absorption effect. We have found through experiments that after the ice bath is removed, the normal temperature stage is about 30 °C. The longer the reaction time, the better the oxidation.

In the subsequent EDS test (Figure 4), the oxygen element on the surface of GO decreased significantly after thermal reduction. The molar ratio of carbon to oxygen decreased from 0.58:1 to 0.41:1, which indicates that thermal reduction removes some oxygen-containing functional groups of GO, which also makes RGM show weak hydrophilicity in subsequent tests.

Figure 5a,b shows that the luminescence spectrum of rGO is a wide peak, that is, the luminescence wavelength is in a large range, and its luminescence depends on the excitation energy. Its emission spectrum and excitation spectrum are symmetrical, which is in line with the luminescence characteristics of organic fluorescent molecules. Figure 5c is the absorption spectrum of rGO dispersion sample tested. In the Figure 5c, the peak of 227 nm comes from the π–π* energy level transition in the C=C bond of *sp*^2^ hybrid, which reflects the increase of *sp*^2^ structure order in go, because part of *sp*^2^ hybrid structure has been restored during the reaction. In addition, there is a weak shoulder peak near 320 nm, corresponding to the electron n–π* transition in C=O, which proves that the oxidation degree of rGO is high [36]. FTIR can show the functional groups on the rGO surface in more detail. It can be seen from the Figure 5d that the rGO surface includes many oxygen-containing groups, including C=O, C-O-C, COOH and O-H, and -C=C-, also has a strong vibration peak, which means that there are many *sp*^2^ clusters inside the rGO.

The above results show that when the emission wavelength is 340 nm, there is an obvious excitation peak at 230 nm, which corresponds to the results of π–π* energy level transition in C=C bond and electron transition in C=O of *sp*^2^ hybrid and corresponds to the energy level shown in the absorption spectrum. In addition, the non-radiative transition of electron hole pair in a small amount of oxygen-containing functional group defects that may be contained in rGO can also excite rGO luminescence.

Figure 6 shows the actual picture of the RGM material in two states. The left picture is the actual photo in the dry state. It can be seen that the rGO is loaded on the non-woven composite membrane in the whole state is black. However, white inclusions can be observed due to the influence of the white non-woven fabric on the color of the membrane. When the RGM is in a wet state, the color becomes darker. The main reason for this phenomenon is that there is a very thin water layer on the surface of the photothermal membrane in the wet state. When the light shines on the surface of the wet photothermal membrane, the light path will be refracted due to the existence of the water layer, that is, the light will shift inward. The light that should have escaped is absorbed, and, finally, the color of the wet photothermal Membrane is deepened [37].

According to the ultraviolet spectrum (Figure 7), the absorption of RGM has a significant change, and greatly enhances its ability to absorb light. The absorbance of RGM composite is about 93% in the wavelength range of 500–2500 nm, which is much higher than the light absorption of non-woven fabrics (8%). The introduction of rGO greatly improved the light absorption performance of the composite material.

The SEM image of the pristine non-woven fabric in Figure 8a shows that the fibers were interspersed in a random network structure. The non-woven fibers are smooth and free of impurities, and the fiber bundle can transport water through capillary action. Loading rGO improves the photothermal conversion efficiency. Magnification shows the rGO sheets are well coated with the fabric. (Figure 8b). Compared with the initial non-woven fabric, the surface of the photothermal membrane after rGO loading and heat treatment is rougher. This rough surface can make the process of multiple reflection of light on its surface, prolong the optical path and, finally, improve the overall light absorption. However, it should be noted that the rGO layer does not block the pores between non-woven fabrics, which are conducive to the escape of steam, which is conducive to the evaporation process of photothermal water.

### 3.2. Light-Induced Evaporation Enhancement via the rGO-Loaded Hydrophilic Nonwoven

To demonstrate the high light absorption of the RGM, water evaporation was performed in four different forms under different light intensities. The water body naturally evaporates, the black non-woven fabric is at the bottom, white, black non-woven fabric on the surface. Through the underlying non-woven core, it looks like the pump to absorb the upper layer of water evaporation.

We can observe that when irradiation with the 1 sun light source for 30 min, the black non-woven fabric is fixed at the bottom of the water container, the evaporation rate is not significantly increased. When the black non-woven fabric is on the top, and the amount of water after evaporation reaches about 0.84 g, four times the black non-woven fabric at the bottom, and two times the white non-woven fabric on the surface (Figure 9).

The results in Figure 9a show that there is little difference between the evaporation rates of non-woven fabric at the top, pure water natural evaporation and RGM at the bottom under one light intensity (1 kW·m^−2^). Natural evaporation of pure water and bottom evaporation of RGM because light acts directly on many water bodies, although the RGM photothermal membrane at the bottom can provide a certain amount of heat for the local water body at the bottom, it cannot heat the overall water volume, so the water evaporation rate under these two evaporation modes is relatively low. Due to its low light absorption (~8%), the non-woven membrane will have obvious reflection and transmission effects on its surface, which makes the membrane surface very limited to convert light into heat. Therefore, even if the non-woven fabric evaporates at the interface, its evaporation flux will not be significantly improved compared with the first two. This is also illustrated by the temperature rise curves of non-woven fabrics and RGM (Figure 9b). The above results show that RGM has better heating performance than previous studies (Figure 9d). Among them, the surface of RGM can be heated up to 71 °C under one light intensity, which is much higher than the heating effect of non-woven fabric under the same conditions (43 °C). In the experiment, the evaporation rate can be estimated simply by the mass loss. The evaporation rate of the above RGM increased with the increase of the radiant light intensity, and the evaporation rate increased from 1.33 kg·m^−2^·h^−1^ to 3.42 kg·m^−2^·h^−1^ (Figure 9c), while the evaporation rate of the other three types is only about 0.28 kg·m^−2^·h^−1^.

The following equation calculated the solar energy conversion efficiency:$$\eta_{eva}=\frac{\dot{m}h_{LV}}{q}$$
where $\eta_{\mathrm{eva}}$ is the evaporation efficiency, *ṁ* represents the stable evaporation rates (kg·m^−2^·h^−1^), *h*_*LV*_ is the total enthalpy of sensible heat and the liquid-vapor phase change (kJ·kg^−1^) and *q* represents the power density of laser irradiation (kW·m^−2^). In our results, the evaporation efficiency is 50.4%. From Figure 9c, it can be shown that the relationship between evaporation efficiency and evaporation rate under different light density of laser indifferent form [38].

It should be noted that when the light intensity increases from one to three, the evaporation rate and photothermal conversion efficiency will not increase in equal proportion. On the contrary, with the increase of light intensity, the increase proportion of evaporation rate decreases. This may be because RGM shows a weak hydrophilic effect (Figure 10), so RGM will not show strong capillarity and water transport capacity like other super hydrophilic materials [39,40,41]. It is this effect that will make the water content on the membrane surface relatively less. Therefore, with the increase of light intensity, the energy obtained by the evaporation system will increase, but the water content on the membrane surface is not enough to support the high-energy state of the system, which will reduce the photothermal conversion efficiency.

To further verify the stability of the rGO loaded on the nonwoven fabric, we sonicated the rGO nonwoven composite membrane for 30 min. From Figure 11, we can see that there is a slight loss of rGO on the beaker after ultrasound, but the change is not significant, which indicates that the stability of rGO loaded on the nonwoven fabric is relatively excellent.

To verify the stability of rGO-loaded nonwoven fabric, we tested the evaporation performance of RGM 10 times under the irradiation of 1 sun light intensity. As shown in Figure 12, we can see that the points in the Figure 12 are unevenly distributed, not in a linear trend. By linear fitting, we can see that the evaporation performance of the entire process material is like that. Near the closing line, it indicates the evaporation mechanical stability of the entire composite membrane is better. The evaporation performance does not increase with amounts of evaporations but has a wide range.

### 3.3. Outdoor Simplified Device Evaporation Experiment

The solar water purification model was simplified, and the experiment was placed on the roof of the Li Yunqiang Experimental building of Hainan University. The rGO-loaded nonwoven fabric has a length of 15 cm, a width of 12 cm and a thickness of 2 mm. It was placed on a simple treated heat-insulating device, which is a foam strip connected by a wire. It was connected by a water-absorbent non-woven fabric in the middle of each foam strip, so that the water at the bottom can be transported to the top through the hydrophilicity of the nonwoven fabric for solar evaporation. The evaporated water condensed into water on a transparent round plastic lid, and then fell by gravity into our collection container. The advantage of this type of evaporation device is simple, and such a set of water evaporation and purification devices can be built at any time in remote areas we recorded the optical density of sunlight from 12:00 to 18:00 outdoors, and the average heat flux is 0.7 kW·m^−2^. In the case of solar radiation, we recorded the surface temperature, air temperature and water volume of the samples separately. From the results in Figure 13, the surface temperature of the sample was obviously changed. When the sunlight irradiates it for about 30 min, the surface temperature of the sample reached 53 °C. Subsequent temperature is somewhat reduced, which may cause a portion of the heat to be removed from the closed system convection, resulting in a decrease in the surface temperature of the sample. Simultaneously, we can also observe from our chart that our water purification is increasing with time. After 6 h, the water volume reached 59 mL. This simple and efficient water purification material combined with the device is suitable for applications in remote areas. It is also worth noting that the bacteria content in the treated seawater is significantly lower than in the original seawater, which indicates the go photothermal membrane has excellent sterilization ability. This is critical for practical applications.

To further prove that the purified water is drinkable, the cytotoxicity of the seawater and the purified water were evaluatedby means of MTT assay on Hela Cells. The results in Figure 14 indicate the purified water have no cell toxicities, while the seawater is highly toxic at high concentrations. This is strong evidence to prove that the method designed is viable for drinking water purification from seawater.

To verify the stability of the materials for seawater desalination, we prepared a 20% sodium chloride solution as a simulating seawater. We conducted a five-day evaporation experiment outdoors. From Figure 15, we can see that the desalination amount of seawater is relatively stable. Figure 15, after evaporation, the surface of the sample leaves a large area of sodium chloride crystals, and the crystals can be washed off with water, which means it can be used repeatedly.

## 4. Conclusions

In summary, the high-photothermal conversion composite material can be produced in a large scale with a low-cost by a physical evaporation phase inversion method in which rGO was loaded into the nonwoven fabric as a main photothermal conversion material. The non-woven fabric is rGO composite membrane, and its mechanical properties are relatively proficient. It can be used repeatedly for seawater purification after sonication based on photothermal heating. The rGO-loaded nonwoven fabric has a high permeability and an evaporation rate of five–six times higher than conventional evaporation. For the solar density of about 726 kw/m^2^, our method can purify 2.35 L of seawater to be evaporated into water vapor for purification. Cell viability studies indicate the purified water has no cell toxicities, while the seawater is highly toxic at high concentrations. Our method designed is viable for drinking water purification from seawater.

## Figures and Tables

**Figure 1 nanomaterials-12-01622-f001:**
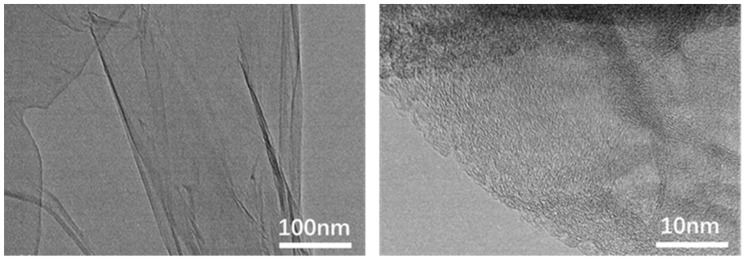
TEM image of GO, On the right is a HRTEM image.

**Figure 2 nanomaterials-12-01622-f002:**
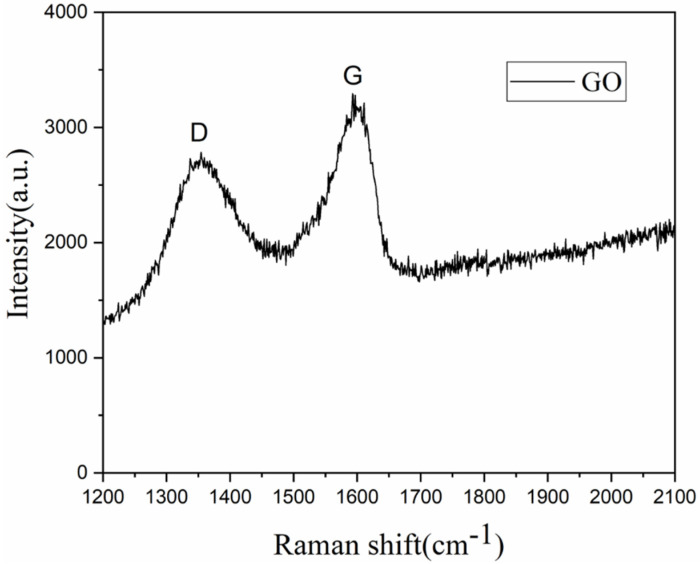
Raman spectra of the prepared GO.

**Figure 3 nanomaterials-12-01622-f003:**
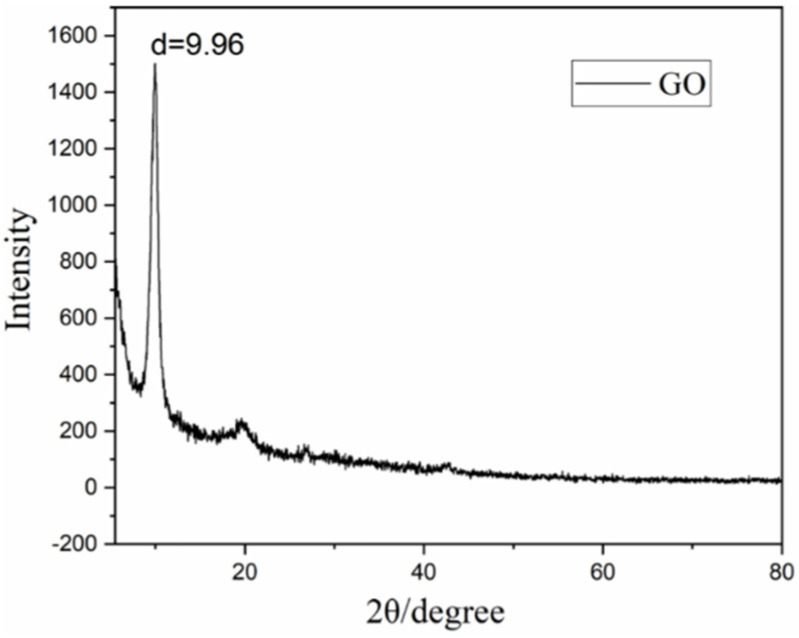
XRD fluorescence spectra of GO.

**Figure 4 nanomaterials-12-01622-f004:**
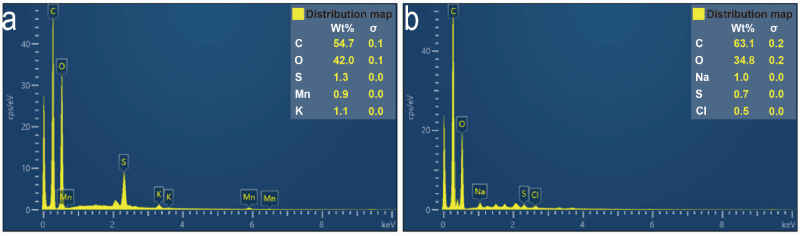
EDS spectra of GO (**a**) and rGO (**b**).

**Figure 5 nanomaterials-12-01622-f005:**
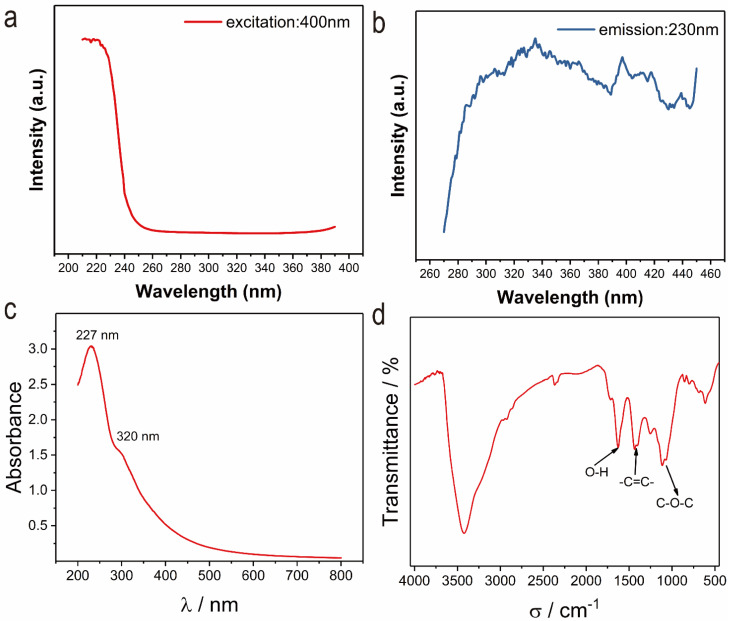
(**a**,**b**) Excitation (Ex: 400 nm) and emission (Em: 230 nm) spectra of rGO. (**c**) UV-Vis absorption spectrum of rGO. (**d**) FTIR spectra of rGO.

**Figure 6 nanomaterials-12-01622-f006:**
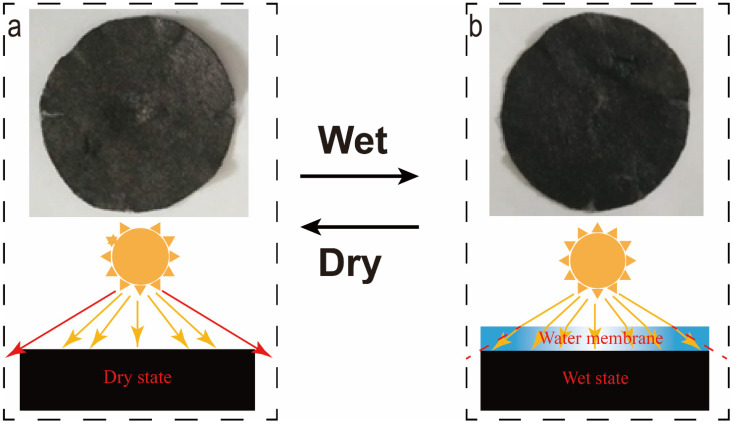
(**a**,**b**) Physical picture of color difference of photothermal membrane under dry and wet conditions (upper part) and corresponding schematic diagram (lower part, the yellow lines indicate absorbed light and red lines indicate escaping light).

**Figure 7 nanomaterials-12-01622-f007:**
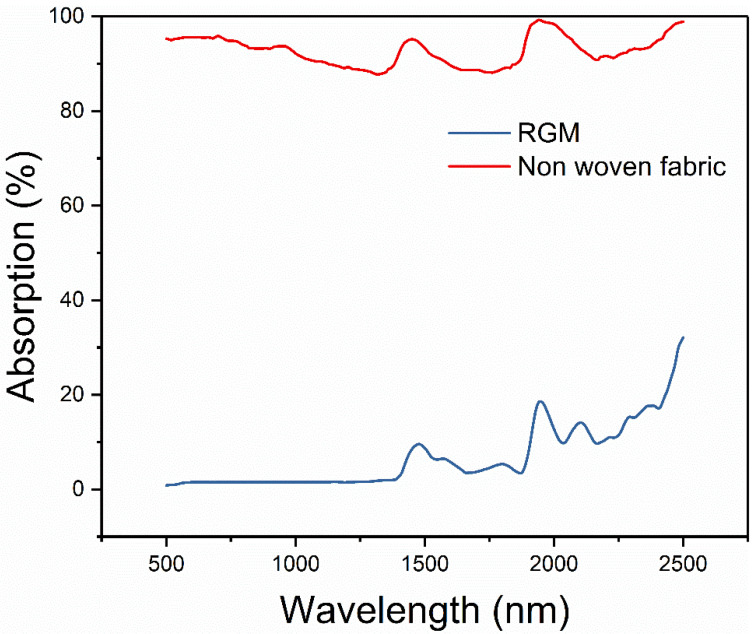
Absorption of RGM and non-woven fabric in the range of 500–2500 nm.

**Figure 8 nanomaterials-12-01622-f008:**
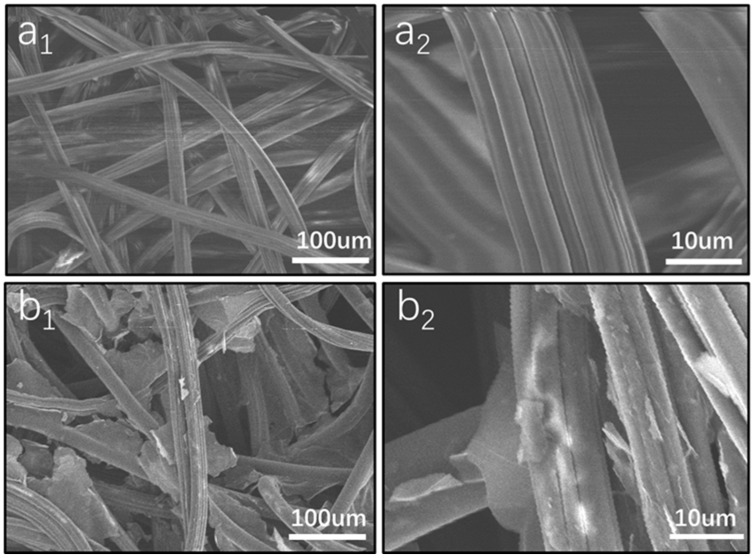
SEM images of the graphene oxide-loaded nonwoven fabric show in different amplified times: (**a1**,**a2**) pristine non-woven fabric of different resolutions; (**b1**,**b2**) modified non-woven fabric with a 0.5% graphene oxide of different resolutions.

**Figure 9 nanomaterials-12-01622-f009:**
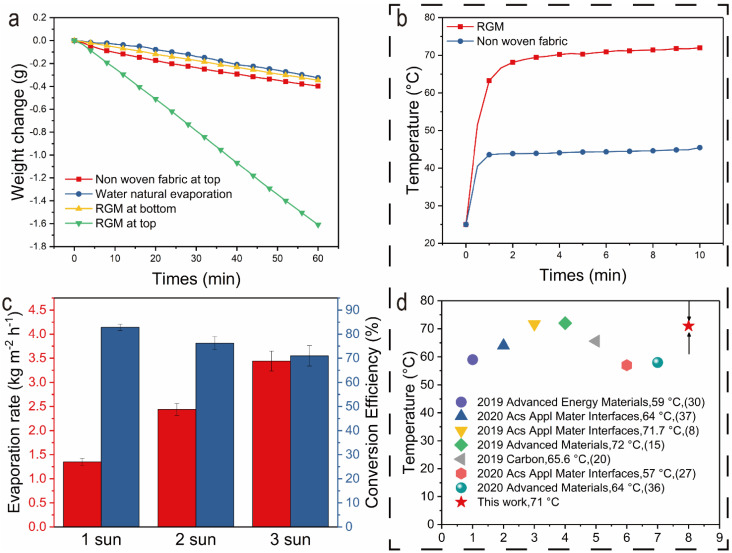
(**a**) Weight change diagram under four evaporation modes (natural evaporation of pure water, evaporation of non-woven fabric at the top, evaporation of RGM at the bottom and evaporation of RGM at the top). (**b**) Temperature rises curve of non-woven fabric and RGM under 1 sun condition. (**c**) Evaporation rate and photothermal conversion efficiency under different light intensities. (**d**) Comparison of heating efficiency in evaporator previously reported (reference numbers are in brackets).

**Figure 10 nanomaterials-12-01622-f010:**
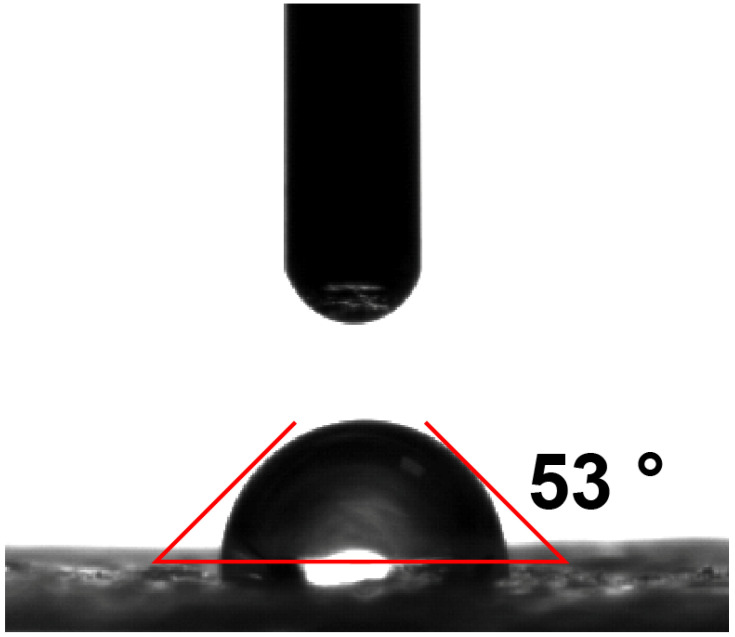
Contact angle test of RGM.

**Figure 11 nanomaterials-12-01622-f011:**
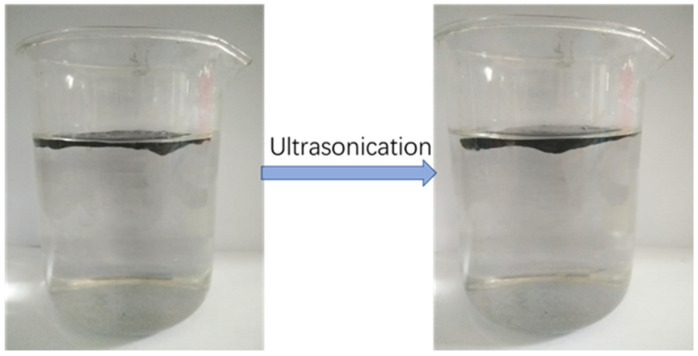
The stability comparison of RGM which are sonicated for 30 min.

**Figure 12 nanomaterials-12-01622-f012:**
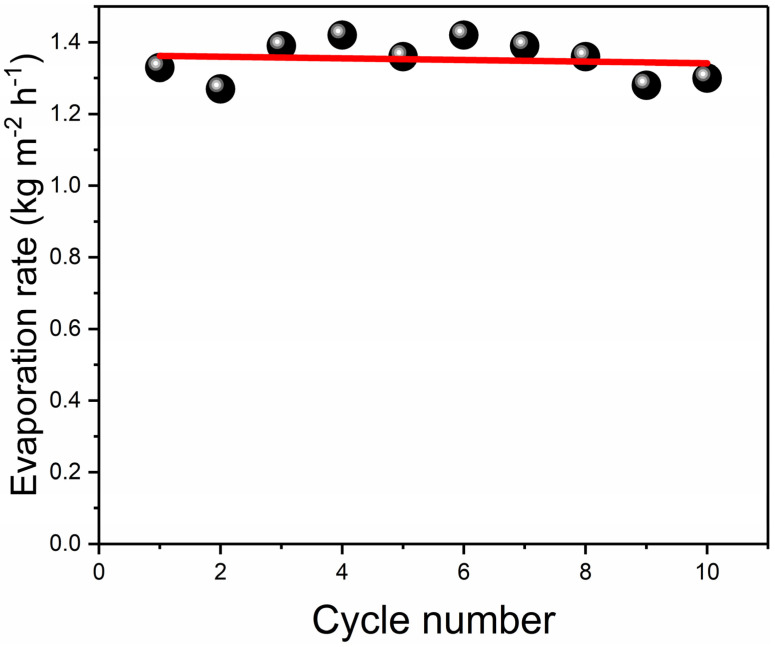
The evaporation rate of RGM on the surface as the function of cycle number.

**Figure 13 nanomaterials-12-01622-f013:**
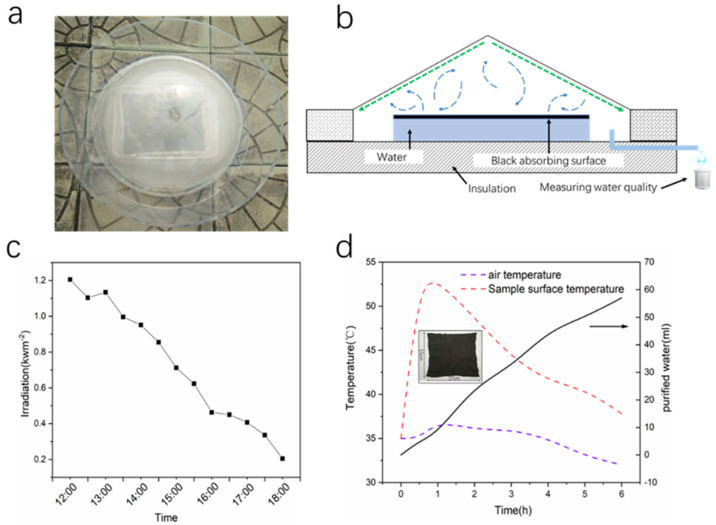
(**a**) A photograph of a large-scale rGO-loaded nonwoven fabric membrane device. (**b**) A schematic diagram of the device collecting water by evaporation. (**c**) Solar radiation recorded over time on a sunny day from 12:00 to 18:00. (**d**) Temperature change with time. The red dotted line represents the surface temperature of the sample, the purple dotted line represents air temperature, the black dotted line represents water evaporation quality changes with time, the illustration shows the size of the sample.

**Figure 14 nanomaterials-12-01622-f014:**
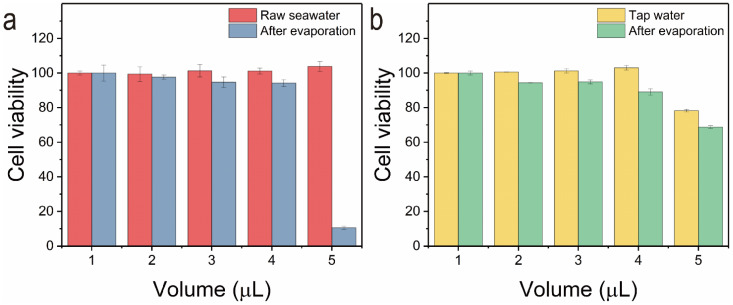
(**a**,**b**) Cell activity before and after seawater and tap water treatment.

**Figure 15 nanomaterials-12-01622-f015:**
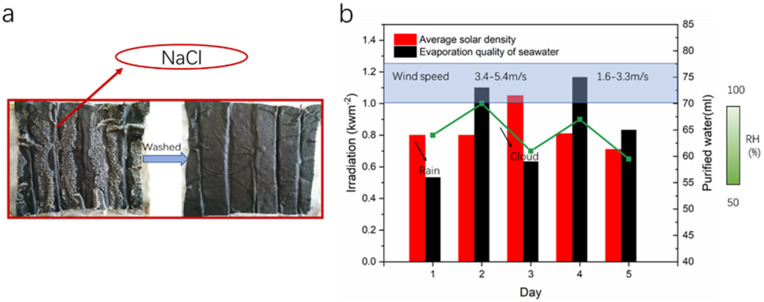
(**a**) A photograph of the surface of the sample after simulating seawater evaporation. (**b**) Cycle days of seawater evaporation. The red column represents the average optical density of sunlight. The black column represents the evaporation quality of water in the afternoon. Green dotted line diagram represents daily air humidity. Blue wireframe represents wind speed.

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
