# Peer review of "Luminescence Reduced Graphene Oxide Based Photothermal Purification of Seawater for Drinkable Purpose"

_nanomaterials, 2022, doi:10.3390/nano12101622_

Round 1
Reviewer 1 Report
The work "Luminescence Reduced Graphene Oxide Based Photothermal Purification of Seawater for Drinkable Purpose" has an actual, important and very interesting subject of the research!
The presented results were interesting and useful.
However the manuscript need some corrections:
- The introduction must be more precisely in the references assigning
(authors give us some large interval, ex.:15-27, 28-34, 37-41);
- The work objective must be more specific and short presented;
- The reference need to write at NANOMATERIALS journal standards;
- The experimental methods description must give some accuracy or/and precision aspects (sure the figures need include the errors bar);
- The discussion section must be amplified (the technical report aspect must change in the article format);
- A comparative reference to literature data should also be included in the discussion section;
- The title of the paper must find several arguments in the discussion section, but also in the conclusions.
Author Response
Reviewer-1
Comments and Suggestions for Authors
The work "Luminescence Reduced Graphene Oxide Based Photothermal Purification of Seawater for Drinkable Purpose" has an actual, important, and very interesting subject of the research!
The presented results were interesting and useful.
However, the manuscript needs some corrections :
- The introduction must be more precisely in the references assigning
(Authors give us some large interval, ex. :15-27, 28-34, 37-41);
Answer: Thank you for your valuable comments. We have revised the scope and area of references to make them more concise.
- The work objective must be more specific and short presented;
Answer: Thank you for your valuable comments. We have deleted the relevant contents of the main body to make the content more concise and specific.
- The reference needs to write at NANOMATERIALS journal standards;
Answer: Thank you for your valuable comments. We have converted the reference format into the standard format of the MDPI official website.
- The experimental methods description must give some accuracy or/and precision aspects (sure the figures need include the errors bar).
Answer :Thank you for your comments. We have revised figures 7 and 12 of the text and added error bars. The figure below is the revised figure.
Fig. 1. Error bar chart.
- The discussion section must be amplified (the technical report aspect must change in the article format).
Answer : Good comments. We have made a more detailed explanations on the color difference of photothermal membrane under dry and wet conditions, the deflection of XRD peak position and its influence on photothermal conversion performance, Raman spectrum, scanning electron microscope and luminescence mechanism in the main body. Please refer to the following contents for details.
(1)
Fig. 2. a,b) Physical picture of color difference of photothermal membrane under dry and wet conditions (upper part) and corresponding schematic diagram (lower part, the yellow lines indicate absorbed light and red lines indicate escaping light).
Above Fig 2 shows the actual picture of the non-woven graphene composite membrane material in two states. The left picture is the actual photo in the dry state. It can be seen the graphene is loaded on the non-woven composite membrane in the whole state is black. However, white inclusions can be observed due to the influence of the white non-woven fabric on the color of the membrane. When the graphene non-woven composite membrane is in a wet state, the color becomes darker. The main reason for this phenomenon is that there is a very thin water layer on the surface of the photothermal membrane in the wet state. When the light shines on the surface of the wet photothermal membrane, the light path will be refracted due to the existence of the water layer, that is, the light will shift inward. The light that should have escaped is absorbed, and finally the color of the wet photothermal membrane is deepened.
(2)
Fig. 3. XRD fluorescence spectra of GO.
As shown in above Fig 3, there is a sharp diffraction peak at 9.96°, slightly lower than the characteristic peak of standard GO centered at about 11°. This may be due to the intercalation of H+ in the go layer during the stripping preparation of GO, which increases the distance between the crystal planes of GO. This expansion layer can aggravate the exposure rate of the edge and improve the overall light absorption effect.
(3)
Fig. 4. Raman spectra of the prepared GO.
Fig 4 is the Raman spectrum of GO. The D peak and the G peak are Raman characteristic peaks of carbon atom crystals, respectively, and their diffraction peaks are centered at1300 cm-1 and 1580 cm-1, respectively. D peak indicates some defects of the carbon atom lattice, and the G peak indicates the in-plane stretching vibration state of the sp2 hybridization of carbon atoms. The D peak and G peak of GO independently developed by the laboratory are at 1359 cm-1 and 1615 cm-1, respectively. ID/IG is the intensity ratio of D peak to G peak. This ratio can refer to the intensity relationship between the two peaks. In the Fig 4, ID/IG is about 0.8, which is less than the generally reported literature value, indicating that the properties of the synthesized GO are closer to the characteristics of graphene, the fewer the defects on its surface and the higher the chemical stability, which will help to improve the stability and service life of the photothermal membrane.
(4)
Fig. 5. SEM images of the graphene-loaded nonwoven fabric show in different amplified times : (a) pristine non-woven fabric; (b) modified non-woven fabric with a 0.5% graphene.
The SEM image of the pristine non-woven fabric in Fig. 5a shows that the fibers were interspersed in a random network structure. The non-woven fibers are smooth and free of impurities, and the fiber bundle can transport water through capillary action. Loading graphene improves the photothermal conversion efficiency. Magnification shows the graphene sheets are well coated with the fabric. (Fig. 5b). Compared with the initial non-woven fabric, the surface of the photothermal membrane after graphene oxide loading and heat treatment is rougher. This rough surface can make the process of multiple reflection of light on its surface, prolong the optical path and finally improve the overall light absorption. However, it should be noted that the rGO layer does not block the pores between non-woven fabrics, which are conducive to the escape of steam, which is conducive to the evaporation process of photothermal water.
- A comparative reference to literature data should also be included in the discussion section.
Answer: Thanks! We have added the comparison diagram of heating performance with other literature in the middle.
Fig. 7. Temperature rises comparison diagram of different documents.
- The title of the paper must find several arguments in the discussion section, but also in the conclusions.
Answer: Thank you for your valuable comments. We have added the elaboration on the luminescence mechanism of rGO in the revision.

Reviewer 2 Report
This paper report the desalination technology using graphene material. Especially, photothermal purification was mentioned and the authors describe that luminescence reduced graphene oxide was used. The manuscript in general is well done and easy to follow. However, as shown below, there are some unclear parts of the manuscript. Thus, if there is an appropriate answer to my major comment and additional data, it can be published.
- Where can I find preparatory procedures or performance results that demonstrate 'luminescence' characteristics? With only the contents described in the text, I am not sure why this material has luminescent properties.
- Graphene oxide? Reduced graphene oxide? In order to make reduced graphene oxide (RGO) in the process of manufacturing graphene flakes using chemical exfoliation, of course, graphene oxide (GO) must first be fabricated. In this manuscript including abstract, those two materials are used interchangeably and confused.
- It is said that the reduction of graphene oxide was done by 180 ℃ for 5 h, but there was no data such as XPS (X-ray photoelectron spectroscopy) or EDS (energy dispersive x-ray spectroscopy) that the reduction occurred well, that is, the oxygen functional group decreased.
Author Response
Reviewer-2
This paper reports the desalination technology using graphene material. Especially, photothermal purification was mentioned and the authors describe that luminescence reduced graphene oxide was used. The manuscript in general is well done and easy to follow. However, as shown below, there are some unclear parts of the manuscript. Thus, if there is an appropriate answer to my major comment and additional data, it can be published.
1.Where can I find preparatory procedures or performance results that demonstrate 'luminescence' characteristics? With only the contents described in the text, I am not sure why this material has luminescent properties.
Answer: Thank you for your comments. To further explain the luminous performance of rGO, we have added RGO and FTIR test and UV-vis-NIR test. The relevant expression results are as follows. These materials have good luminescence, however, in this paper, it is not our major work, so we did not describe too much about it.
Fig. 8. a,b) Excitation(Ex:400nm) and emission(Em:230nm) spectra of rGO. c) UV-Vis absorption spectrum of rGO. d) FTIR spectra of rGO.
Fig. 8a and Fig. 8b show that the luminescence spectrum of rGO is a wide peak, that is, the luminescence wavelength is in a large range, and its luminescence depends on the excitation energy. Its emission spectrum and excitation spectrum are symmetrical, which is in line with the luminescence characteristics of organic fluorescent molecules. Fig. 8c is the absorption spectrum of rGO dispersion sample tested. In the Fig. 8c, the peak of 227 nm comes from the π - π* energy level transition in the C = C bond of sp2 hybrid, which reflects the increase of sp2 structure order in go, because part of sp2 hybrid structure has been restored during the reaction. In addition, there is a weak shoulder peak near 320 nm, corresponding to the electron n - π * transition in C = O, which proves that the oxidation degree of rGO is high. FTIR can show the functional groups on the rGO surface in more detail. It can be seen from the Fig. 8d that the rGO surface includes many oxygen-containing groups, including C = O, C-O-C, COOH and O-H, and -C = C- also has a strong vibration peak, which means that there are many sp2 clusters inside the rGO.
The above results show that when the emission wavelength is 340 nm, there is an obvious excitation peak at 230 nm, which corresponds to the results of π - π * energy level transition in C = C bond and electron transition in C = O of sp2 hybrid and corresponds to the energy level shown in the absorption spectrum. In addition, the non-radiative transition of electron hole pair in a small amount of oxygen-containing functional group defects that may be contained in rGO can also excite rGO luminescence.
2.Graphene oxide? Reduced graphene oxide? To make reduced graphene oxide (RGO) in the process of manufacturing graphene flakes using chemical exfoliation, of course, graphene oxide (GO) must first be fabricated. In this manuscript including abstract, those two materials are used interchangeably and confused.
Answer :Thank you for your valuable comments on the full text. In the process of proofreading the full text, we found that there was confusion in the use of relevant proper nouns in the article. The revised full text has unified all relevant proper nouns, making the full text more rigorous and scientific. For relevant revisions, please refer to the full text of the latest edition.
3.It is said that the reduction of graphene oxide was done by 180 ℃ for 5 h, but there was no data such as XPS (X-ray photoelectron spectroscopy) or EDS (energy dispersive x-ray spectroscopy) that the reduction occurred well, that is, the oxygen functional group decreased.
Answer :
Fig. 9. EDS spectra of GO(a) and rGO(b).
Thank you for your valuable comments. In the revised version, we have carried out EDS test on GO and rGO after heat treatment. As shown in the Fig. 9, there are obvious differences in oxygen elements between GO and rGO. The molar ratio of carbon to oxygen in GO is about 0.58:1, while rGO is about 0.41:1, which shows that heat treatment reduction can significantly reduce the oxygen-containing functional groups in GO.

Round 2
Reviewer 1 Report
The revised manuscript "Luminescence Reduced Graphene Oxide Based Photothermal Purification of Seawater for Drinkable Purpose" respond at suggestions, corrections and recommendations.
However, the references must write at NANOMATERIALS journal standards.
